# Clinical Performance of Diagnostic Methods in Third Molar Teeth with Early Occlusal Caries

**DOI:** 10.3390/diagnostics13020284

**Published:** 2023-01-12

**Authors:** Nazan Kocak-Topbas, Kıvanç Kamburoğlu, Ayşe Tuğba Ertürk-Avunduk, Mehmet Ozgur Ozemre, Nejlan Eratam, Esra Ece Çakmak

**Affiliations:** 1Department of Dentomaxillofacial Radiology, Faculty of Dentistry, Mersin University, Mersin 33343, Turkey; 2Department of Dentomaxillofacial Radiology, Faculty of Dentistry, Ankara University, Ankara 06110, Turkey; 3Department of Restorative Dentistry, Faculty of Dentistry, Mersin University, Mersin 33343, Turkey

**Keywords:** laser fluorescence, near-infrared light transillumination, Micro-CT, non-cavitated occlusal caries, visual analog scale

## Abstract

The aim of this study was to compare the diagnostic performance of clinical visual examination (ICDAS II), digital periapical radiography (PR), near infrared light transillumination (NIR-LT), and laser fluorescence (LF) to microcomputed tomography (Micro-CT) which is the reference standard for the detection of non-cavitated occlusal enamel and dentin caries in third molar teeth. Potential participants were consecutively recruited. In this prospective study, 101 third molars of 101 patients were examined; the molars had non-cavitated occlusal caries which required extraction. ICDAS II, PR, NIR-LT and LF examinations were carried out by two blinded examiners. Reference standard was determined by micro-CT imaging seven days after extraction. Accuracy rate, sensitivity, specificity, predictive values and areas under receiver operating characteristic (ROC) curves (Az) were statistically analyzed. Nonparametric variables were subjected to the Kruskal–Wallis Test. Significance level was set as *p* < 0.05. NIR-LT had the highest sensitivity (99.67–99.67%) and accuracy (78.22–77.23%) for the determination of occlusal enamel caries according to examiners 1 and 2, respectively. LF method had the highest sensitivity (70.83–54.17%) and accuracy (66.34–59.41%) for determining occlusal dentin caries according to examiners 1 and 2, respectively. The ROC curve (Az) value ranged between 0.524 and 0.726 for the different methods tested. Most effective methods for the diagnosis of occlusal enamel and dentin caries were determined to be NIR-LT and LF pen methods, respectively. The present prospective clinical study showed that NIR-LT and LF-Pen were a reliable modality for the detection of occlusal enamel and dentin caries without ionizing radiation.

## 1. Introduction

Risk assessment and early diagnosis of adult caries and application of conservative and/or restorative treatment is essential [1]. Occlusal fissural caries is frequently observed in routine clinical practice; however, diagnosis may be problematic due to masking effect of healthy enamel around the caries [2]. Conventional techniques are useful especially in the detection of cavitated occlusal caries lesions, whereas supplementary techniques are usually needed in the detection of non-cavitated hidden caries lesions [1,3,4]. Conventional techniques are visual inspection, probing (Nyvad system, ICDAS I and II, Univiss System) and radiographic examination by using CCD (charge coupled device), CMOS (complementary metal oxide semiconductors) and PSP (photostimulable phosphor plate) systems. Fluorescent techniques, namely laser fluorescent (DIAGNOdent pen) and quantitative light fluorescent (QLF) are other clinical techniques used for occlusal caries diagnosis. Enhanced visual techniques which can be used for occlusal caries diagnosis are fiber optic transillumination (FOTI), digital fiber optic transillumination (DIFOTI), and near infrared light transillumination (NIR-LT) [5].

Ideal diagnostic method for occlusal caries detection should be non-destructive, simple, fast, accurate, easily applicable, and cost effective. It should also be able to show the depth and extend of the caries lesion in order to enable early detection and early conservative intervention [3]. Visual inspection is the simplest primary method utilized in order to obtain immediate diagnosis. It is without extra costs; however, it may be insufficient in the diagnosis of initial occlusal caries lesions [6,7]. In addition, visual inspection reveals high specificity at cost of low sensitivity, reliability and repeatability [8,9,10,11]. International Caries Detection and Assessment System (ICDAS II) is a visual system that defines caries intensity in 6 levels [9]. Visual inspection should be supported by additional diagnostic techniques. For radiological assessment, digital periapical radiography (PR) is frequently used to determine caries depth extending into the dentine [10,11,12,13]. In addition, bitewing radiographies are used for radiologic assessment.

Today, it has been shown that fluorescence techniques and improved visual techniques supported by clinical studies are effective in the early diagnosis of occlusal caries, therefore it is no wonder that their popularity is increasing by the day [11]. Porphyrins are responsible from fluorescence level released from cariogenic bacteria in caries lesions when exposed to 655 nm red light [13]. Fluorescence intensity is an indicator of caries depth and the DIAGNOdent Pen device is utilized as a commercially available laser fluorescence (LF) method. Near infrared light transillumination method is mainly based on developments in digital fiber optic transillumination and use of near infrared light [14,15]. Instead of cold visible light, NIR-LT uses invisible near infrared light with longer wavelengths, which decreases light scattering effects and enables deeper light penetration into enamel and dentine [16,17,18]. 

Microcomputed tomography (Micro-CT) is a non-invasive and non-destructive innovative technique that is used to visualize tooth structures in three dimensions with very high accuracy. Micro-CT replaced microscopic evaluation and is now considered as the gold standard method for *in-vitro* caries assessment [12,19]. Recently, micro-CT was used to measure mineral concentration of enamel and dentine and to assess cortical bone, caries cavities and root canal treatment ability of materials in vitro [12,18,19]. Ultrahigh effective doses along with radiation exposure, long scan times and high costs preclude its clinical use [20].

The majority of the studies in the previous literature regarding comparison of examination techniques in the detection of caries were carried out under in vitro conditions due to ethical concerns. However, it is not always an acceptable approach to defend the clinical applicability of the evaluation performed under in vitro conditions.

Therefore, the aim of this study was to compare and evaluate the diagnostic accuracy of the initial occlusal caries lesions in third molars requiring extraction for different indications by using ICDAS II, PR, NIR-LT and LF methods under in vivo conditions with the gold standard micro-CT method. In addition, clinical application time and patient comfort analysis were also conducted for each method. The null hypothesis of the study was that the clinical performance of the evaluated methods in detecting non-cavitated occlusal caries was not statistically significantly different for caries at the enamel and dentin level at different depths.

## 2. Materials and Methods

This prospective clinical research protocol was approved by Mersin University Clinical Ethics Research Council in Turkey (Approval Code: No. 2016/322; Approval Date: 6 October 2016). Data was collected at Faculty of Dentistry, Mersin University. In this clinical trial, STARD statements (Standards for Reporting of Diagnostic Accuracy Studies) were followed (Figure 1) and the study protocol was registered on www.clinicaltrial.gov (Identifier: NCT05064566).

### 2.1. Selection Criteria

#### 2.1.1. Inclusion Criteria

Only ASA status 1 (according to the American Anesthesiologists Association) healthy patients were included. The following inclusion criteria were used in the present research: (1) fully erupted permanent dentition, (2) absence of fixed prosthetic restoration, restorative material and orthodontic apparatus, (3) the patients aged at least 18 years, (4) at least one erupted third molar of the patient (third mandibular or third maxillary molar from each patient was included), (5) non-cavitated occlusal caries in the third molars, (6) absence of hypoplasia or hypomineralization of third molars, (7) third molars that required extraction for surgical, orthodontic and prosthetic reasons. Patients were asked to participate in this additional examination if clinical signs indicated an existing caries risk or activity and additional diagnosis was deemed necessary.

#### 2.1.2. Exclusion Criteria

Patients with ASA status 2–6 (according to the American Society of Anesthesiologists) who were not healthy were excluded from the study. The following exclusion criteria were used in this research: (1) completely non-erupted permanent dentition, (2) fixed orthodontic apparatus, prosthetic restoration and filling material, (3) a maximum age of 18 years, (4) at least one third non-erupted molar teeth (non-erupted third mandibular or non-erupted third maxillary molar from each patient was excluded), (5) cavitated occlusal caries, (6) teeth with hypoplasia or hypomineralization, (7) third molars that have dental occlusion, function, caries-free and not required to be extracted for other reasons were excluded from the study.

Sample size was calculated by using the G*Power test software. The study sample size was calculated as 40 patients considering that a small effect size of d = 0.614, an alpha level 0.05, a target power of 80% and two predictors were expected in the final model. 

Potential participants were consecutively recruited from December 2020 to April 2021, and they all provided informed consent for participation. In this prospective cohort study, 140 patients were examined; however, the final sample comprised of 101 third molars of 101 patients with non-cavitated occlusal caries requiring extraction in the sample. A total of 39 patients/third molars were excluded since patients did not show up for their extraction appointments or extensive damage occurred in their dental crowns during extraction. Finally, 101 patients with a mean age of 30.80 years (standard deviation 9.70) agreed to participate in the overall study. The final sample comprised of 101 third molars requiring extraction with non-cavitated but suspected occlusal caries from 101 patients. This study was divided into three sections as shown in Figure 1. A pre-extraction (clinical visual examination, PR, NIR-LT, and LF) and post-extraction assessment (micro-CT) of each tooth were performed. 

Each of the *in-vivo* parts (N.K.T, A.T.E.A) and *in-vitro* micro-CT parts (N.E, E.C) of the study were conducted by two blinded independent observers who were not aware of the characteristics of teeth they observed in advance. A total of four different observers were employed in the in vivo and in vitro parts of the study and were unaware of each other’s examination results.

A benchmarking examiner (K.K.) trained each of the observers by using 15 occlusal surfaces; however, no calibration procedures were performed. Afterwards, these teeth were not included in the study.

For each clinical method, clinical procedural application time and comfort/pain analysis by using Visual Analog Scale (VAS) were evaluated by consensus. Clinical procedural application times for total time (from the beginning to the end of the procedure) were measured by using a stopwatch for each method. In addition, for each clinical technique, patients were also asked to score their feelings (in relation to comfort and pain) from absence of discomfort and pain to extreme discomfort and pain (between 0 and 10). 

### 2.2. Clinical Visual Examination (ICDAS II)

Clinical visual examination of teeth was conducted under standard conditions by two observers [21]. According to the International Caries Detection and Assessment System (ICDAS II), score lesion severity was recorded [22]. The utilized criteria were as follows: (0) sound surfaces, (1) first observable variations in the enamel, (2) distinct observable variations in the enamel, (3) localized enamel breakdown, and (4) underlying dark shadow reflecting from dentin. Diagnosis of enamel caries was conducted by using Codes of 1, 2, and 3, and for dentin caries diagnosis a code of 4 was utilized. 

### 2.3. Examination by Using Intraoral Radiography Method

Digital intraoral periapical radiographs were exposed by using an X-ray unit (ProX^®^, Planmeca, Helsinki, Finland) operated at 60 kVp and 7 mA with a size 2 (3 × 4 cm) photo-stimulable phosphor plate (PSP) detector (Planmeca, Helsinki, Finland). Image recording was set at 35μm pixel size and 16-bit color. Imaging was performed using standardized paralleling technique equipment with circular collimation (Rinn Manufacturing Company, Elgin, IL, USA) with a focus receptor distance of 20 cm and an image exposure time of 0.08 s and scanned with (Proscanner^®^, Planmeca, Inc., Helsinki, Finland), stored and assessed by using built-in Romexis software (Planmeca, Inc., Helsinki, Finland). Pulpal root canal, dentine and enamel visibility were used as indicators of optimal image quality, determined by consensus. The presence of occlusal enamel or dentin caries was determined by using the classification suggested by Schaefer et al. [23] as follows: (0) sound surfaces/no radiolucency visible, (1) radiolucency in the outer half of the enamel, (2) radiolucency in the inner half of the enamel, (3) radiolucency restricted to the outer half of the dentin, (4) radiolucency restricted to the inner half of the dentin. For enamel caries, codes of 1 or 2 were used, and for dentin caries, codes of 3 and 4 were used.

### 2.4. Examination by Using NIR-LT Method

NIR-LT method was performed on air-dried teeth with the use of DIAGNO cam device (KaVo, Biberach, Germany) laser type 1 (780). The camera was focused on the occlusal surface at the correct focus-to-object distance and appropriate images were saved in the KID software (KaVo Integrated Desktop/version 2.4.1.6374). The stages of occlusal enamel and dentin caries were determined by using the criteria suggested by Schaefer et al. [23] as follows: (0) sound/no visible changes in the enamel or no less translucent spots, (1) first or established caries lesion restricted to enamel; no visible translucent dentin, (2) less translucent dentin. The diagnosis of enamel caries was coded as 1 and diagnosis of dentine caries was coded as 2.

### 2.5. Examination by Using LF Method

The fissure probe type 2 DIAGNOdent Pen (KaVo, Biberach, Germany) was applied to the designated location and performed with rotational movements in order to examine all fissure walls to record the maximum value. Values between 0 and 99 were recorded for the examined tooth. Sound areas ranged between 0 and 13. Occlusal enamel caries was associated with values between 14 and 29 and dentin caries were associated with values ≥ 29 [24,25].

### 2.6. Validation by Using Micro-CT Examination/Reference Standard Method

Micro-CT has been accepted as the gold standard method over time, replacing microscopic methods for in vitro caries assessment, as it provides reproducible results with high sensitivity and very high accuracy in tooth structures [12,19]. After extraction, teeth were stored in 100% humidity distilled water. On the 7th day after teeth extraction, a PET/Micro-CT (Super Aargus: Sedecal USA Inc., Madrid, Spain) device was used for gold standard micro-CT imaging. Images were obtained by using 68 9 68 mm FOV, 40 kVp and 140 mA with 0.06 voxel size. The irradiation time was 10 min 30 s. After 3D reconstruction by using Amide software (version 1.0.4), coronal, sagittal, and axial planes were obtained. The validation of both enamel and dentin caries was performed by consensus of 2 examiners (N.E, E.C) by using data viewer. All teeth were qualified to one of five groups according to the scale proposed by Luczaj-Cepowicz et al. [26] as follows: (0) intact, (2) E1, outer half of enamel radiolucent, (3) E2, inner half of enamel radiolucent, (4) D1, outer third of dentine radiolucent, (5) D2, inner two thirds of dentine radiolucent. Caries was considered to be advanced as either in enamel or dentin. Carious lesions with E1, E2, D1 and D2 radiological progression level in enamel were considered positive, whereas 0 was considered to be intact. Radiological progression level of D1 and D2 for lesions extending into dentin was considered to be diseased, whereas 0, E1, and E2 were thought to be intact. 

After 1-week interval time, 45 patients were recalled in order to calculate the intra- and inter-observer reliability of each diagnostic method. These patients also consisted of patients who came to their appointments for other dental treatments. Evaluation of all examination methods was repeated independently. Figure 2 shows representative images obtained by using all examination methods for dentin and enamel.

### 2.7. Statistical Analysis

The clinical performance of different diagnostic methods was based on the accuracy rate, sensitivity (SE) (positivity in caries), specificity (SP) (negativity in caries), negative predictive value (NPV), and positive predictive value (PPV). The receiver operating characteristic (ROC) curves (Az) were calculated to show the distinguishing ability of a test [27,28]. Variables that were not normally distributed (nonparametric) were evaluated by Kruskal–Wallis Test. The multiple comparison relationship between the methods in terms of duration of clinical application and patient comfort was evaluated by using the Mann–Whitney U test. Intra- and inter-observer reproducibility were assessed by intraclass correlation coefficient (ICC) [29]. Significance level was set at *p* < 0.05. 

## 3. Results

Regarding the micro-CT examination, it was determined that from a total of 101 occlusal sites, 26 were intact (0), 6 had carious lesions in enamel outer half (E1), 21 had carious lesions located at the inner half of the enamel (E2), 21 had carious lesions located at dentin outer one third (D1), and 27 sites had carious lesions at dentin middle 1/3 (D2). Table 1 shows SE, SP, PPV, NPV, accuracy rates, and areas under the ROC curve/Az values for enamel and dentin caries for each diagnostic method. 

NIR-LT had the highest SE (99.67–99.67%) and accuracy (78.22–77.23%) for determining enamel caries, while PR had the lowest SE (58.67–61.33%) and accuracy (62.38–61.39%) according to examiners 1 and 2, respectively. The LF (76–76%) and clinical visual examination (82.67–82.67%) methods had intermediate SE values for determining enamel caries according to examiners 1 and 2, respectively. LF method had the highest SE (70.83–54.17%) and accuracy (66.34–59.41%) for determining dentin caries, while clinical visual examination had the lowest SE (10.42–18.75%) and accuracy (54.46–57.43%) according to examiners 1 and 2, respectively. The NIR-LT (64.58–52.08%) and PR (33.33–33.33%) methods had intermediate SE values for determining dentin caries according to examiners 1 and 2, respectively. All diagnostic methods showed lower diagnostic performance when compared to the gold standard for diagnosing occlusal enamel and dentin caries with ROC curve (Az) values between 0.524 and 0.726. 

For each diagnostic technique, ICC was calculated from both readings of measurements between each observer (intra-observer) and repeated measurements among all observers (inter-observer). The ICC values changed from 0 (no agreement) to 1 (perfect agreement). High correlation was accepted for an ICC value of >0.75, good correlation was accepted for 0.74 > ICC > 0.60 value, fair correlation was accepted for 0.59 > ICC > 0.40 value and poor correlation was accepted for ICC value of less than 0.40. The ICC values showed a perfect consistency, indicating a high correlation for the clinical diagnostic methods within and between examiners (Table 2). 

The consistency of two examiners and each clinical diagnostic method with the micro-CT method showed fair to good correlation (Table 3).

When the methods were listed from the most comfortable to the most uncomfortable, clinical visual examination was followed by the LF pen method, NIR-LT and PR technique (Table 4). Regarding the clinical application time, it was determined that the quickest method was the clinical visual examination followed by the LF pen method, NIR-LT and PR (Table 4). The patient comfort score and clinical application time of the clinical visual examination group were lower than those of the PR group (*p* < 0.05). The clinical application time of the clinical visual examination group was lower than that of the LF group (*p* < 0.05). The patient comfort score and clinical application time of the clinical visual examination group were lower than those of the NIR-LT group (*p* < 0.05). The patient comfort score and clinical application time of the PR group were higher than those of the LF group (*p* < 0.05). The clinical application time of the PR group was higher than that of the NIR-LT group (*p* < 0.05). The patient comfort score and clinical application time of the LF group were lower than those of the NIR-LT group (*p* < 0.05) (Table 4).

## 4. Discussion

Today, occlusal caries assessment and detection plays a vital role in routine dental clinical practice and is a prerequisite for appropriate treatment. In general, the stage in which caries can be diagnosed both visually and radiologically is the dentin demineralization stage. The size and boundaries of deep dentin caries can be easily determined by traditional methods. Due to the different morphological structures of the occlusal surfaces of the teeth, early detection of occlusal caries without cavitation by visual inspection may be very difficult. 

New techniques (NIR-LT and LF method) utilized for the diagnosis of early occlusal caries in third molar teeth might be more successful compared to conventional methods (clinical visual examination and PR) in terms of diagnosis, application time and patient comfort. Therefore, the teeth sample included in the present study had no cavitation, their occlusal surfaces were intact and possible caries were at the initial level.

This hypothesis of the present research was that all methods would reveal similar diagnostic results. However, it was determined that the NIR-LT method for enamel caries and the LF method for dentin caries were superior to other diagnostic methods in comparison to gold standard micro-CT technique. These results contradict the previous studies that attributed a significant benefit to radiography for the detection of caries on occlusal surfaces [30,31,32]. On the other hand, the study findings were in line with the results of other studies which concluded that PR as an additional method had no significant benefit when used to analyze occlusal surfaces only [33,34]. This finding could be attributable to superposition of the enamel tubercles on PR and malposition of the third molar teeth for clinical visual examination which might reduce the visibility and diagnosis of caries.

The use of micro-CT showed a significant jump in dental research including the validation of in vitro caries detection in permanent teeth [35,36]. The use of micro-CT as a gold standard in caries diagnosis was evaluated by Soviero et al. [20]. They showed that the findings obtained by using histological analysis and micro-CT examination were comparable. Advantage of micro-CT over histological examination was that there was no tissue damage during radiological scanning. The sample of this study consisted of only third molar teeth, because in vitro micro-CT, which was the gold standard method, could only be used on extracted teeth and was compared with all clinical methods. For this reason, third molars of patients with space limitations and with other clinical complaints were included in the study as they were frequently extracted. The micro-CT findings of this study showed that caries lesions extended deeper than expected on micro-CT images despite the selection of clinically non-cavitated teeth in the sample.

It was determined that clinical application of NIR-LT proved to be the most effective method in the detection of occlusal enamel caries which was indiscernible with the use PR assessment in most cases. In a previous in vivo study, visual inspection examiners observed most of the diagnosed caries lesions on occlusal surfaces in young adults [23]. PR and NIR-LT showed a limited benefit that might be due to sample selection and design of the mentioned study. However, authors suggested that NIR-LT might be preferred over X-ray-based methods in clinical practice due to radiation concerns and clinical practicability. Authors also suggested the use of PR in clinical situations where multiple deep caries lesions were diagnosed. This kind of diagnostic strategy might be helpful to avoid false diagnosis and to determine optimal preventive and restorative dental care based on an optimal diagnostic evaluation [23]. Unlike occlusal caries lesions, proximal caries lesions were difficult to detect and diagnose via visual clinical inspection due to their location and clinical difficulties [37,38]. It was determined that LF pen method was superior to other methods in detecting occlusal dentin caries. In this regard, it was concluded that it was not an acceptable practice to consider radiography the first-choice diagnostic method without clinical examination. In addition, ICC values obtained for four different in vivo methods were quite high, suggesting excellent repeatability for the diagnostic methods utilized in the presented study. However, it was determined that there were fair results for the consistency of the two examiners and each clinical diagnostic method with the gold standard micro-CT method (Table 3). 

To the best of our knowledge, no previous comprehensive study compared several techniques in terms of caries diagnostic ability and clinical application time and comfort/pain analysis (VAS). However, few studies evaluated these parameters for PR radiographic techniques [39,40]. The authors detected a significant difference in image acquisition time between patients with discomfort and those with no discomfort (*p* < 0.001) for PSP sensor [39]. Another study recommended phosphor plate to be used for children with mild level of discomfort [40].

It should be noted that during periapical radiographic imaging of third molar teeth, malposition and posterior position of the teeth along with gagging reflex may cause longer times to be required in order to obtain the ideal image and high scoring for patient comfort (VAS scale).

According to the study findings, rankings for discomfort and clinical application time were identical. The difference between all methods in terms of clinical application time was statistically significant (*p* < 0.001). In terms of patient comfort, the difference was statistically significant, except for clinical visual examination and LF and digital PR and NIR-LT relations (*p* < 0.001). When the methods were listed from the most comfortable one to the most uncomfortable, clinical visual examination was followed by the LF pen method, NIR-LT and PR technique. In regard to clinical application time, it was determined that the quickest method was the clinical visual examination followed by the LF pen method, NIR-LT and PR. The study findings suggested that patient comfort has a positive impact on shorter clinical application time.

The ICCMS radiography classification is a method that determines the prognosis with a minimally invasive approach based on the activity of the caries and the caries risk of the patient. A very good connection can be established with the ICDAS classification in terms of diagnosis and treatment [41]. However, the third molars, which constitute the sample of the study, consisted of teeth that required extraction as treatment planning. Therefore, the radiographic method used by Schaefer et al. and Kühnisch et al. [37] was assumed to be more appropriate in regard to present research design.

In the present study, apart from the fact that clinical visual examination was a simple, fast, and comfortable method in the detection of occlusal caries, once again its relatively high compliance and reliability were emphasized compared to other methods and the gold standard. 

A meta-analysis in the literature revealed that analog or digital radiographs are sensitive enough to detect early enamel caries. It was stated that CBCT showed superior sensitivity to analog or digital radiographs. However, it was not recommended for routine caries detection due to its limited applicability in practice, high radiation dose, and potential for caries-like artifacts from existing restorations [42]. For this reason, CBCT was not used as an index test in this study. In addition, it was stated in the study that the time elapsed between the index tests and the reference test should ideally be less than three months [42]. In the present study, this period was determined as 1 week.

In a meta-analysis conducted by Macey et al. in 2020, the authors determined that there was no statistically significant difference in the accuracy of red, blue, or green fluorescent-based devices. In the present study, a red fluorescence-based device, the DIAGNOdent pen was used. The specified meta-analysis consisted of in vitro-weighted publications. [43]. Until a more accurate reference standard is developed which prioritizes in vivo safe use, the evaluation of such studies will be possible with the application of micro-CT after extraction, as in this study. It is thought that with the developments in 3D technology, concerns about the intraoral applicability of the reference standard will become absolete [43]. 

Optical coherence tomography (OCT) creates cross-sectional images of tooth structure by reflection and backscattering of light in an attempt to measure optical reflection with depth recording [44]. A systematic review by Macey et al. in 2021 showed that Optical Coherence Tomography (OCT) is superior to NIR and fiber optic technologies for the detection of early enamel caries. OCT is not currently available to general practitioners, but it is expected to be included in routine dental examinations in light of higher quality research and development. Transillumination and OCT devices were determiend to be valuable tools in the detection of enamel caries [44]. Similarly, in the presented study, NIR-LT was determined to be more successful in detecting enamel caries in comparison to other methods. Authors of a review article suggested that the possible design of an ideal diagnostic test accuracy study was the one in which participants required tooth extraction were identified, allowing index testing to be conducted in a clinical setting, and histology for the reference standard after tooth extraction was possible [44]. In line with the mentioned review paper, extracted teeth were assessed before and after extraction by different radiological methods, micro-CT being the reference standard.

In a meta-analysis in which diagnostic studies of proximal surface caries were systematically reviewed, authors reported that in vitro studies were frequently included in the literature (120), whereas clinical studies were in the minority (9). Bitewing radiography was determined to be more successful in detecting proximal caries [45]. It was also stated that an acceptable reference standard should be developed for clinical caries detection and diagnostic studies. Obviously, in our notion, different reference standards developed in 3D technology would bring a new dimension to clinical occlusal caries research.

In a similar study conducted by Taşsöker et al. [46] in 2020, the researchers determined the reference standard as histology and evaluated the performance of ICDAS 2, NIR-LT, LF Pen in detecting carious lesions in third molars without cavitation. The authors determined that NIR-LT was more effective in diagnosing caries. In line with the mentioned study, this study also determined that NIR-LT was more successful and accurate in detecting occlusal enamel caries in comparison to other techniques assessed.

In the study conducted by Mortensen et al. in 2018, which determined the reference standard as ICDAS and evaluated the clinical performance of impedance spectroscopy, laser fluorescence and bitewing radiographs in detecting occlusal caries in premolar and molar teeth without fillings, the LF pen was determined to be especially useful in the absence of bitewing radiographs or when ionizing radiation should be avoided. The authors suggested that it might be used as a second opinion to visual inspection. However, in the present study, LF pen was determined to be effective in detecting occlusal dentin caries.

Today, while the pandemic is slowing down due to the COVID-19 vaccine, dental clinics with high patient circulation are considered as high-risk areas for transmission. In addition, another important problem in routine clinics is the dental treatment expectations of COVID-19-positive patients [47]. The use of air and water sprays which produce visible aerosols during dental examination are an important factor for contamination [47]. In this study, clinical visual examination of caries may be considered as a risk factor when compared to other methods in this sense. Although other methods do not produce aerosols, the probability of contamination can be higher than that of clinical visual examination in terms of longer clinical application times. It has been reported that the optimum solution to all these problems is the use of an air cleaner system, which proved to be effective in the literature. In addition to protective equipment, an air cleaner system consisting of a HEPA 14 filter capable of cleaning the air on a larger scale than a normal air cleaner would be effective in reducing both air pollution and contamination. There is a need for further controlled experimental studies using the inspection methods and portable air cleaner system describes in the presented study in order to reduce the microbiological risk.

Early intervention of caries is of paramount importance as treatment delayed due to misdiagnosis may lead to tooth loss. The resulting cavities are potential candidates for implant prosthetic rehabilitation [48]. It is believed that the more successful detection of enamel and dentin caries with NIR-LT and LF pen in the early period will play an important role in avoiding more costly implant operations.

With the bad changes in eating habits, the possibility of having diabetes is increasing day by day in addition to the formation of caries. As is known, this pathology affects the oral microbiome, vascularization and healing processes, and salivary composition [49]. As a multifactorial disease, the high prevalence of dental caries in patients with type 1 diabetes, and especially in patients with poor metabolic control, may be due to the interaction of genetic factors, oral cariogenic bacteria, food intake, and oral hygiene. First, insulin deficiency may cause degenerative changes in the salivary glands and lead to a decrease in salivary flow and salivary buffer capacity. Low-quality diets may adversely affect oral health in individuals with type 1 diabetes through their effects on immune function and glycemic control [50]. In this context, caries diagnosis and treatment should be conducted under optimal conditions in the early period and with appropriate clinical examination methods, especially in uncompromised diabetes patients; potential tooth loss and therefore healing disorder and the risk of infection can be prevented in this way.

The observers examined third molars. It is known that the occlusal surfaces are very different from other teeth in terms of morphology, quantity of enamel and dentin, and position in the mouth. The frequent occurrence of variations observed in the morphology of the occlusal surfaces in the study sample, and the fact that the majority of the sample consisted of malposed teeth might be a possible limitation of this study. In addition, only occlusal surfaces were evaluated as they were difficult to diagnose. However, advanced methods (especially NIR-LT and LF) analyzed for caries detection may also be used for interproximal caries which were out of this study’s scope. Interproximal caries results may be different from those obtained by this study. Another limitation of this study was that different pain thresholds of the patients were not taken into account. The pain thresholds measured by the digital pressure algometer and the responses of the patients according to the VAS scale were not evaluated. However, in the present study, the relative comfort of the methods was prioritized rather than the pain threshold of the patients. In addition, caries risk assessment form data that could not be obtained and analyzed from all patients who participated in the study due to clinical intensity.

## 5. Conclusions

ICC values calculated for different clinic diagnostic methods showed high repeatability. This prospective clinical study showed that NIR-LT and LF-Pen are reliable modalities for the detection of occlusal enamel and dentin caries without using ionizing radiation. The most comfortable method with the shortest application time is the clinical visual examination method, whereas the most uncomfortable method with the longest application time is the PR method. 

## Figures and Tables

**Figure 1 diagnostics-13-00284-f001:**
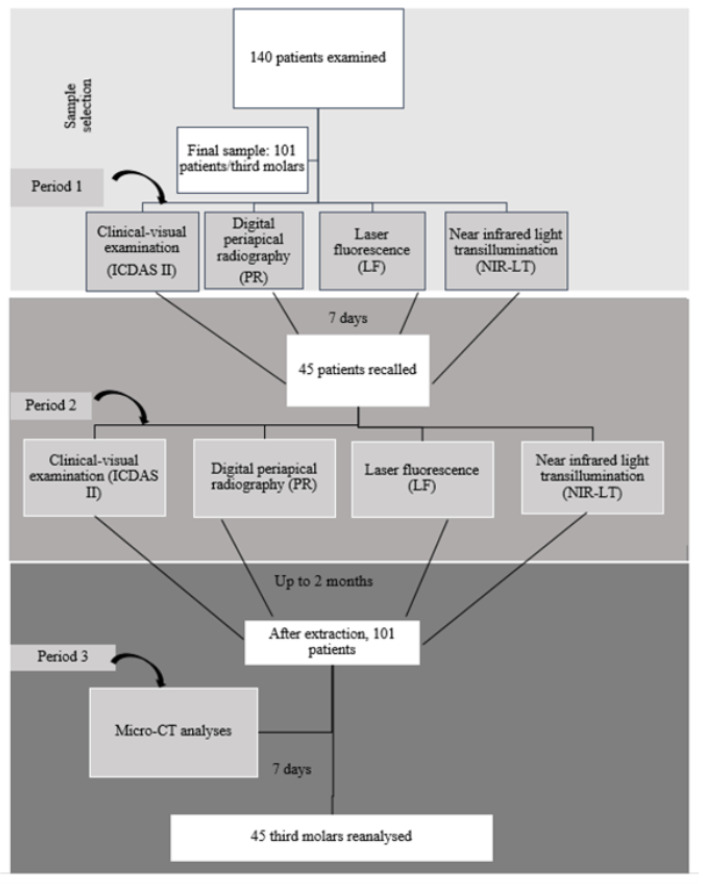
Study flowchart demonstrating the sequence of examinations performed in the study.

**Figure 2 diagnostics-13-00284-f002:**
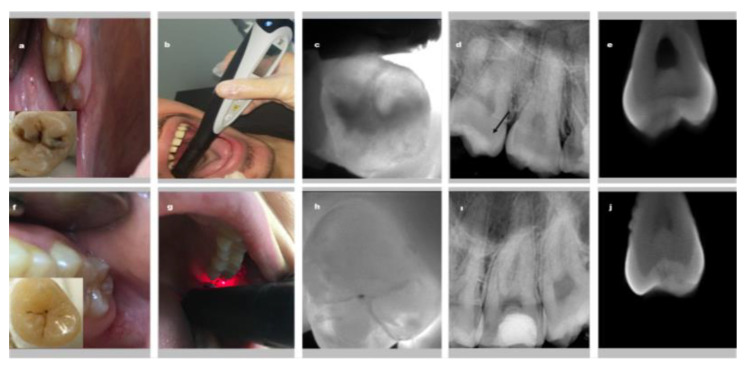
Representative images demonstrating all examination methods for dentin (**a**–**f**) and enamel (**f**–**j**) caries; (**a**–**f**) clinical visual examination, (**b**–**g**) laser fluorescence (**c**–**h**) near infrared light transillumination, (**d**–**i**) periapical radiography (the black arrow shows dentin caries in figure d), (**e**–**j**) micro-CT.

**Table 1 diagnostics-13-00284-t001:** Sensitivity, specificity, positive predictive value, negative predictive value and accuracy of clinical visual examination, periapical radiography (PR), laser fluorescence (LF) and near-infrared light transillumination (NIR-LT) in comparison to micro-CT at the enamel and dentin thresholds.

	Method	Examiner	Sensitivity (%)	Specificity (%)	Positive Predictive Value (%)	Negative Predictive Value (%)	Area under the ROC Curve/Az	Accuracy (%)	*p*
Enamel	Clinical visual examination	Examiner 1	(82.67)	(57.69)	(84.93)	(53.57)	0.702	(76.24)	<0.001
	Clinical visual examination	Examiner 2	(82.67)	(57.69)	(84.93)	(53.57)	0.702	(76.24)	<0.001
	PR	Examiner 1	(58.67)	(73.08)	(86.27)	(38.00)	0.659	(62.38)	<0.005
	PR	Examiner 2	(61.33)	(61.54)	(82.14)	(35.56)	0.614	(61.39)	<0.043
	LF	Examiner 1	(76.00)	(69.23)	(87.69)	(50.00)	0.726	(74.26)	<0.001
	LF	Examiner 2	(76.00)	(65.38)	(86.36)	(48.57)	0.707	(73.27)	<0.001
	NIR-LT	Examiner 1	(90.67)	(38.46)	(80.95)	(58.82)	0.646	(77.23)	<0.001
	NIR-LT	Examiner 2	(90.67)	(42.31)	(81.93)	(61.11)	0.665	(78.22)	<0.001
Dentin	Clinical visual examination	Examiner 1	(10.42)	(94.34)	(62.50)	(53.76)	0.524	(54.46)	0.377
	Clinical visual examination	Examiner 2	(18.75)	(92.45)	(69.23)	(55.68)	0.556	(57.43)	0.093
	PR	Examiner 1	(33.33)	(88.68)	(72.73)	(59.49)	0.610	(62.38)	<0.007
	PR	Examiner 2	(33.33)	(86.79)	(69.57)	(58.97)	0.601	(61.39)	<0.016
	LF	Examiner 1	(70.83)	(62.26)	(62.96)	(70.21)	0.665	(66.34)	<0.001
	LF	Examiner 2	(54.17)	(64.15)	(57.78)	(60.71)	0.592	(59.41)	0.064
	NIR-LT	Examiner 1	(64.58)	(66.04)	(63.27)	(67.31)	0.653	(65.35)	<0.002
	NIR-LT	Examiner 2	(52.08)	(62.26)	(55.56)	(58.93)	0.572	(57.43)	0.147

LF: laser fluorescence; NIR-LT: near-infrared light transillumination; PR: periapical radiography; ROC: receiver operating characteristic; Az: ROC Area index. *p* < 0.05 was statistically significant.

**Table 2 diagnostics-13-00284-t002:** ICC values for intra- and -inter examiner variability each diagnostic method tested.

ICC	Examiner 1	Examiner 2	Inter-Examiner
Clinical visual examination	0.963 (0.932–0.979)	0.978 (0.960–0.988)	0.967 (0.952–0.978)
PR	0.950 (0.910–0.973)	0.968 (0.942–0.982)	0.982 (0.974–0.988)
LF	0.964 (0.935–0.980)	0.940 (0.890–0.967)	0.965 (0.947–0.976)
NIR-LT	0.918 (0.851–0.955)	0.930 (0.873–0.962)	0.949 (0.925–0.966)

ICC: intraclass correlation coefficient; LF: laser fluorescence; NIR-LT: near-infrared light transillumination; PR: periapical radiography.

**Table 3 diagnostics-13-00284-t003:** Comparison of the consistency of two examiners and each diagnostic method with the micro-CT method.

ICC	Micro-CT- Examiner 1	Micro-CT- Examiner 2
Clinical visual examination	0.617 (0.432–0.742)	0.613 (0.425–0.739)
PR	0.563 (0.351–0.705)	0.530 (0.303–0.683)
LF	0.542 (0.320–0.691)	0.484 (0.235–0.652)
NIR-LT	0.507 (0.269–0.668)	0.426 (0.148–0.613)

ICC: intraclass correlation coefficient; LF; laser fluorescence; NIR-LT: near-infrared light transillumination; PR: periapical radiography.

**Table 4 diagnostics-13-00284-t004:** Comparison of patient comfort score and clinical application time in seconds for each diagnostic test as mean, median, standard deviation. (s.d), *p* value.

^1^	Patient Comfort Score			Clinical Application Time		
	Mean	s.d.	Median	Mean	s.d.	Median
Clinical visual examination	0.33	±0.60	0.00	3.96	±2.07	4.00
PR	4.06	±3.47	4.00	55.16	±3.18	55.00
LF	0.39	±0.58	0.00	9.68	±4.89	8.00
NIR-LT	3.11	±2.40	3.00	17.14	±8.12	15.00
^2^	Patient comfort score (*p* value)			Clinical application time (*p* value)		
Clinical visual examination—PR	<0.001			<0.001		
Clinical visual examination—LF	0.292			<0.001		
Clinical visual examination—NIR-LT	<0.001			<0.001		
PR—LF	<0.001			<0.001		
PR—NIR-LT	0.092			<0.001		
LF—NIR-LT	<0.001			<0.001		

^1^ Kruskal–Wallis Test. ^2^ Mann–Whitney U Test. LF: laser fluorescence; NIR-LT: near-infrared light transillumination; PR: periapical radiography. *p* < 0.05 was statistically significant.

## Data Availability

Not applicable.

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
