# Peer review of "Clinical Performance of Diagnostic Methods in Third Molar Teeth with Early Occlusal Caries"

_diagnostics, 2023, doi:10.3390/diagnostics13020284_

Round 1

Reviewer 1 Report

1-    Why only third molars were included from the whole sample?

2-    How were the observers/clinicians calibrated for the procedures

3-    Patients have different pain threshold so how was this considered when recording the rate of their feeling.

4-    Were the patients recalled just for the study?

Author Response

Reviewer 1 suggestions:

  • Why only third molars were included from the whole sample?

Answer: Thank you. The article has been reviewed and arranged in accordance with your suggestions (see p 9, line 322-326).

  • How were the observers/clinicians calibrated for the procedures?

Answer: Thank you. The article has been reviewed and arranged in accordance with your suggestions (see p 4, line 143-149).

  • Patients have different pain threshold so how was this considered when recording the rate of their feeling.

Answer: Thank you. The article has been reviewed and arranged in accordance with your suggestions ( see p 12, line 472-476).

  • Were the patients recalled just for the study?

Answer: Thank you. The article has been reviewed and arranged in accordance with your suggestions (see p 6, line 217-218).

Reviewer 2 Report

Very interesting study and a well presented paper. The authors do offer a comparable insight into the performance of various diagnostic approaches for detecting dental caries. A number of items, however, need to be addressed:

- Details of examiner calibration aren't clear (some mention later but pre-examination training and calibration details are missing) This is important given a relatively fair operator consistency noted;

- When, post-extraction, was MicroCTexamination conducted?

- Was there a caries-risk assessment completed? This data would've been very helpful;

- Is there a reason why the ICCMS radiograph classification wasn't used? Would've linked rather well with the ICDAS classification and diagnosis;

- The discussion doesn't quite emphasise the excellent findings of the simple yet reliable clinical examination;

- There is no discussion re several important papers and comparing/contrasting of findings: Walsh, T., Macey, R., Riley, P., Glenny, A. M., Schwendicke, F., Worthington, H. V., ... & Sengupta, A. (2021). Imaging modalities to inform the detection and diagnosis of early caries. Cochrane Database of Systematic Reviews, (3). Macey, R., Walsh, T., Riley, P., Glenny, A. M., Worthington, H. V., Fee, P. A., ... & Ricketts, D. (2020). Fluorescence devices for the detection of dental caries. Cochrane Database of Systematic Reviews, (12). Macey, R., Walsh, T., Riley, P., Hogan, R., Glenny, A. M., Worthington, H. V., ... & Ricketts, D. (2021). Transillumination and optical coherence tomography for the detection and diagnosis of enamel caries. Cochrane Database of Systematic Reviews, (1).Janjic Rankovic, M., Kapor, S., Khazaei, Y., Crispin, A., Schüler, I., Krause, F., ... & Kühnisch, J. (2021). Systematic review and meta-analysis of diagnostic studies of proximal surface caries. Clinical oral investigations25(11), 6069-6079.

Author Response

  • Details of examiner calibration aren't clear (some mention later but pre-examination training and calibration details are missing) This is important given a relatively fair operator consistency noted;

Answer: Thank you. The article has been reviewed and arranged in accordance with your suggestions ( see p 4, line 143-149).

  • When, post-extraction, was MicroCTexamination conducted?

Answer: Thank you. The sentence specified has been arranged in accordance with your suggestions; see p 6, line 202.

  • Was there a caries-risk assessment completed? This data would've been very helpful; Answer: Thank you. The article has been reviewed and arranged in accordance with your suggestions ( see p 12, line 476-478).
  • Is there a reason why the ICCMS radiograph classification wasn't used? Would've linked rather well with the ICDAS classification and diagnosis;

Answer: Thank you. The article has been reviewed and arranged in accordance with your suggestions ( see p 10, line 370-376).

  • The discussion doesn't quite emphasise the excellent findings of the simple yet reliable clinical examination;

Answer: Thank you. The specified sentence was added as suggested; see page 10, line 377-380.

  • There is no discussion re several important papers and comparing/contrasting of findings: Walsh, T., Macey, R., Riley, P., Glenny, A. M., Schwendicke, F., Worthington, H. V., ... & Sengupta, A. (2021). Imaging modalities to inform the detection and diagnosis of early caries. Cochrane Database of Systematic Reviews,

Answer: Thank you. The article has been reviewed and arranged in accordance with your suggestions ( see p 11, line 397-411).

  • There is no discussion re several important papers and comparing/contrasting of findings: Macey, R., Walsh, T., Riley, P., Glenny, A. M., Worthington, H. V., Fee, P. A., ... & Ricketts, D. (2020). Fluorescence devices for the detection of dental caries. Cochrane Database of Systematic Reviews,

Answer: Thank you. The article has been reviewed and arranged in accordance with your suggestions ( see p 11, line 389-396).

  • There is no discussion re several important papers and comparing/contrasting of findings: Macey, R., Walsh, T., Riley, P., Hogan, R., Glenny, A. M., Worthington, H. V., ... & Ricketts, D. (2021). Transillumination and optical coherence tomography for the detection and diagnosis of enamel caries. Cochrane Database of Systematic Reviews,

Answer: Thank you. The article has been reviewed and arranged in accordance with your suggestions ( see p 11, line 397-411).

  • There is no discussion reseveral important papers and comparing/contrasting of findings: Janjic Rankovic, M., Kapor, S., Khazaei, Y., Crispin, A., Schüler, I., Krause, F., ... & Kühnisch, J. (2021). Systematic review and meta-analysis of diagnostic studies of proximal surface caries. Clinical oral investigations, 25(11), 6069-6079.

Answer: Thank you. The article has been reviewed and arranged in accordance with your suggestions ( see p 11, line 412-418).

Reviewer 3 Report

Dear authors,

this manuscript is interesting and well-conducted. The methods applied for studying the accuracy in diagnosing methods and the statistical analysis are appropriate and for these reasons I think that the article can be accepted. However, I suggested some revisions in order to improve the quality and the clarity of your work.

Title: please add to the title ... in third molar teeth

Please in the abstract and all over the text, talk in third person about the study and use a more passive verbs. For example, I would change "Our aim" at the beginning of the abstract and write "The aim of this study was..."

Line 18: take care of the grammar: "..., however;" 
Always at that point, be more concise. Readers would know how many teeth you analyzed (the final number of the sample) and not also the initial sample.

Line 20: the two examiners were blinded? Please specify it

Please define "SE" in the abstract or erase it and insert with the correct and complete words.

Line 29: Erase "For the first time". Please talk only about your study in the abstract, then in the discussion or in the conclusion you can tell this

Line 57: erase the ,

Line 59-60: also bitewing radiographies are used for radiologic assessment

Line 62: please insert at least a phrase to introduce the speech about fluorescent techniques to detect caries

Line 100: Please divide the "selection criteria" into inclusion and exclusion criteria

Line 106: Please correct the grammar "reasons, If"

Line 119, 121: Our study... Our final sample, please be unpersonal here and all over the text

Line 124: Two observers. I didn't understand if the two observers that evaluated the MicroCT exams and those that performed the clinical examinations are blinded. It is not clear if they are aware or not of the teeth they observed. If they are aware of this fact, you have to insert it in the limits of the study. In addition, it seems that two operators examined the MicroCT and other two made the clinical exams. Please write in materials and methods that different observes were employed for these tasks

Figure 2: please provide better images, with high resolution and low distortions

Lines 288-289: you can erase this phrase. You've just told it in the previous sections of the manuscript

You discussed your results in a proper way in the discussion section. However, a section of limits of your study is lacking. I would add that you analyse third molars that are so different from the other teeth in terms of morphology, quantity of enamel and dentin, position in the mouth.
Even it is obvious, I would specify the reason way the VAS scale resulted higher for periapical radiography of third molars.
In addition, you selectioned teeth for caries in the occlusal surface but the methods you analyzed for detecting caries (especially NIR and LF) can also be used for interproximal caries that you do not evaluated and results may be different from those obtained by you. Other limits can also be mentioned 

Moreover I would add a little paragraph where you talk about the clinical significance of your study.

Author Response

  • Title: please add to the title ... in third molar teeth

Answer: Thank you. The title was added as suggested; see page 1, line 2-3.

  • Please in the abstract and all over the text, talk in third person about the study and use a more passive verbs. For example, I would change "Our aim" at the beginning of the abstract and write "The aim of this study was..."

Answer: Thank you. The entire article has been reviewed and arranged in accordance with your suggestions (for example see p 1, line 14, 30).

  • Line 18: take care of the grammar: "..., however;" Always at that point, be more concise. Readers would know how many teeth you analyzed (the final number of the sample) and not also the initial sample.

Answer: Thank you. The sentence specified has been arranged in accordance with your suggestions; see p 1, line 18-20.

  • Line 20: the two examiners were blinded? Please specify it

Answer: Thank you. The sentence specified has been arranged in accordance with your suggestions; see p 1, line 20-21.

  • Please define "SE" in the abstract or erase it and insert with the correct and complete words.

Answer: Thank you. The abbreviation specified has been removed in accordance with your suggestions and has added correct and complete words; see p 1, line 24-27.

  • Line 29: Erase "For the first time". Please talk only about your study in the abstract, then in the discussion or in the conclusion you can tell this

Answer: Thank you. The specified adverbial phrase has been removed in accordance with your suggestions; see p 1, line 30-32.

  • Line 57: erase the ,

Answer: Thank you. The desired change has been made in accordance with your suggestions; see p 2, line 57.

  • Line 59-60: also bitewing radiographies are used for radiologic assessment

Answer: Thank you. The specified sentence has been added and arranged in accordance with your suggestions; see p 2, line 61-62.

  • Line 62: please insert at least a phrase to introduce the speech about fluorescent techniques to detect caries

Answer: Thank you. The requested sentence has been added and arranged in accordance with your suggestions; see p 2, line 63-65.

  • Line 100: Please divide the "selection criteria" into inclusion and exclusion criteria

Answer: Thank you. The requested paragraph has been divided and arranged in accordance with your suggestions; see p 3-4, line 103-127.

  • Line 106: Please correct the grammar "reasons, If"

Answer: Thank you. The sentence has been rearranged in accordance with your suggestions; see p 3, line 111-114.

  • Line 119, 121: Our study... Our final sample, please be unpersonal here and all over the text

Answer: Thank you. The sentence and all over the text has been rearranged in accordance with your suggestions; see p 4, line 132-133.

  • Line 124: Two observers. I didn't understand if the two observers that evaluated the MicroCT exams and those that performed the clinical examinations are blinded. It is not clear if they are aware or not of the teeth they observed. If they are aware of this fact, you have to insert it in the limits of the study. In addition, it seems that two operators examined the MicroCT and other two made the clinical exams. Please write in materials and methods that different observes were employed for these tasks

Answer: Thank you. The article has been reviewed and arranged in accordance with your suggestions; see p 4, line 143-146.

  • Figure 2: please provide better images, with high resolution and low distortions

Answer: Thank you. The figure 2 has been rearranged in accordance with your suggestions; see p 6, line 221.

  • Lines 288-289: you can erase this phrase. You've just told it in the previous sections of the manuscript

Answer: Thank you. This phrase has been erased and arranged in accordance with your suggestions; see p 9, line 318.

  • You discussed your results in a proper way in the discussion section. However, a section of limits of your study is lacking. I would add that you analyse third molars that are so different from the other teeth in terms of morphology, quantity of enamel and dentin, position in the mouth.

Answer: Thank you. The limitation paragraph has been added and arranged in accordance with your suggestions; see p 12, line 465-469.

  • Even it is obvious, I would specify the reason way the VAS scale resulted higher for periapical radiography of third molars.

Answer: Thank you. This paragraph has been added and arranged in accordance with your suggestions; see p 10, line 356-358.

  • In addition, you selectioned teeth for caries in the occlusal surface but the methods you analyzed for detecting caries (especially NIR and LF) can also be used for interproximal caries that you do not evaluated and results may be different from those obtained by you. Other limits can also be mentioned.

Answer: Thank you. This paragraph has been added and arranged in accordance with your suggestions; see p 12, line 464-468.

  • Moreover I would add a little paragraph where you talk about the clinical significance of your study.

Answer: Thank you. This paragraph has been added and arranged in accordance with your suggestions; see p 9, line 301-306.

Reviewer 4 Report

The paper is very interesting. However, before acceptance, authors should point out some aspects about prognosis of caries and the potential impact on covid 19 pandemic. In particular:

1) How the different methods can impact on potential contamination of sars-cov2 in the light of usage of prevention systems like air purifiers? Please cite DOI10.3390/ijerph19095139

2) Can the caries have an impact on the future outcome of potential implant-prosthetic rehabilitations, when a destruptive lesion can bring to the loss of tooth? and such methods are useful? Please cite DOI10.23805/JO.2018.10.04.04

3) what about such topic on dibaetes patients? they are at risk of infection. Consider the possibility to evaluate the treatment of caries also in the light of potential loss of teeth. Cite DOI10.3390/ijerph191811735

Author Response

  • How the different methods can impact on potential contamination of sars-cov2 in the light of usage of prevention systems like air purifiers? Please cite DOI10.3390/ijerph19095139

Answer: Thank you. One paragraph has been added and arranged in accordance with your suggestions; see p 11-12, line 431-445.

  • Can the caries have an impact on the future outcome of potential implant-prosthetic rehabilitations, when a destruptive lesion can bring to the loss of tooth? and such methods are useful? Please cite DOI10.23805/JO.2018.10.04.04

Answer: Thank you. One paragraph has been added and arranged in accordance with your suggestions; see p 12, line 446-450.

  • What about such topic on dibaetes patients? they are at risk of infection. Consider the possibility to evaluate the treatment of caries also in the light of potential loss of teeth. Cite DOI10.3390/ijerph191811735

Answer: Thank you. One paragraph has been added and arranged in accordance with your suggestions; see p 12, line 451-463.

Round 2

Reviewer 2 Report

Thank you for addressing my comments. 

Reviewer 3 Report

I do not have any other relevant comment for the authors. Thanks for the revision